# An offline-first electronic health record for vulnerable populations: A mixed-methods feasibility study

Henry Ashista[1,2]*, Alanis Santiago Comas[1], Taylor Selby[3], Mohammad Yasir Essar[4], Jude Alawa[5], Samar Al-Hajj[6], Erica Nelson[7]

**1** Hikma Health, San Jose, California, United States of America, **2** Department of Emergency Medicine, Alameda County Health System, Oakland, California, United States of America, **3** Ohio University Heritage College of Osteopathic Medicine, Dublin, Ohio, United States of America, **4** Kabul University of Medical Sciences, Kabul, Afghanistan, **5** Department of Orthopedic Surgery, Hospital for Special Surgery, New York, New York, United States of America, **6** Department of Epidemiology and Population Health, Faculty of Health Sciences, American University of Beirut, Beirut, Lebanon, **7** Harvard Medical School, Department of Emergency Medicine, Boston, Massachusetts, United States of America

* hcashwor@gmail.com

## Abstract

While there is a growing body of research showing that Electronic Health Records (EHRs) can improve health outcomes in limited resourced settings, these EHRs often require continual access to an internet connection and are challenging to customize and deploy. To address these challenges, Hikma Health (HH) has designed a free, open-source and offline-first EHR for physicians providing care in low-resource settings. This paper describes a mixed-methods study to understand the feasibility of the HH EHR in two clinics that care for displaced or rural populations in Lebanon and Nicaragua. Clinic demographics and metrics were collected through REDCap and descriptive statistics were analyzed. Using a framework analysis, in-depth interviews were conducted at both sites until thematic saturation. Interviews were coded by three authors with an inter-rater reliability kappa score of 7.6. Quantitative data showed that after about three hours of training and three weeks of use, participants were comfortable using the HH EHR and the HH EHR decreased patient interview times by three minutes. Quantitative data showed that these findings were due to the ease and simplicity of the modular workflows. However, when used in settings that required syncing multiple times during a patient's encounter with different clinicians, the system faced challenges due to inconsistent network connectivity and the design of the sync functionality. The HH EHR was a feasible solution for offline-first settings. Both clinical sites observed that the implementation of the EHR enhanced documentation, decreased medical errors, and improved patient outcomes through tracking medications and diseases.

**Data availability statement:** The data used is of a sensitive nature and therefore cannot be shared openly under the IRB protocols. The datasets used and/or analysed during the current study are available from Hikma Health (info@hikmahealth) on reasonable request.

**Funding:** Hikma Health funded the necessary tools to conduct this research project including Dictaphones, software subscriptions, and transcription services. No members of the research team were paid by Hikma Health. No member of the Hikma Health Board played a specific role in the conceptualization, design, data collection, analysis, decision to publish, or preparation of the manuscript. Organizational volunteers HA and AS were responsible for the conceptualization, design, data collection, analysis, decision to publish, and the preparation of the manuscript along with all other authors.

**Competing interests:** We have read the journal's policy and the authors of this manuscript have the following competing interests: Authors HA and AS are both volunteers for Hikma Health (HH). They are not compensated financially or otherwise for their work at Hikma Health which is a 501c3 nonprofit organization. All other authors have no conflicts of interest to state. No members of the Hikma Health board played any part in the data collection, analysis, interpretation, or reporting.

## Author summary

Research has shown that electronic health records help improve patient care and health systems management. However, electronic health records are often expensive, require continuous internet connectivity, and are not designed by clinicians. This makes implementing electronic health records in limited resource settings challenging and prevents their use in caring for patients who need them most. In this paper, we studied the implementation of a free, open-sourced, offline-first electronic health record that doesn't require continuous internet activity across two limited-resource clinics in Lebanon and Nicaragua. We found that an off-line first electronic health record was feasible and helped improve both quantitative clinic metrics as well as clinicians' views of their ability to care for patients. We also found that an offline-first electronic health record presents challenges when clinics require continuous syncing to the cloud throughout a patient's encounter. More innovation is needed to help develop electronic health records in these settings. When developed in line with human-centric design processes, specifically sgeared towards use, such EHRs help improve the care provided to often displaced and migratory populations.

## Introduction

Globally, more than 100 million people were forcibly displaced in 2022, seeking refuge from armed conflicts, natural disasters, extreme poverty, and torture [1,2]. According to the United Nations High Commissioner for Refugees, approximately 84% of forcibly displaced individuals flee to low- and middle-income countries (LMICs) with developing health systems and insufficient resources to provide services to this vulnerable population [2]. Due to insufficient resources in host countries, sustainable access to healthcare for displaced communities is often limited [3–5].

LMICs face many challenges in providing appropriate healthcare to rural and displaced populations. Among these challenges are the maintenance of patient health records and the provision of consistent and timely access to quality healthcare, particularly for chronic health conditions [6]. Electronic health record (EHR) systems provide an immense opportunity to improve access to healthcare and health outcomes for displaced populations by providing consolidated, secure, and accurate patient information and medical records, which are essential for guiding diagnostic and treatment decision-making [6,7]. However, the development of EHR system solutions for low-resource settings and vulnerable populations remains scarce, largely due to funding, data security, an the difficulties of implementing and modifying such systems across multiple stake holders (clinics, software developers, and governments) to meet the specific needs of particular populations [8].

Previous literature has highlighted the significance of EHRs in improving accessibility to quality healthcare when adopted [8,9]. Studies suggest that EHRs serve as a tool for improving communication, documentation, and adherence to guidelines,

which all contribute to better health outcomes [8–13]. A study by Khader et al. examined health outcomes and complications of Palestinian refugees with diabetes mellitus over a three-year period. Their findings suggest that using an EHR decreased the morbidity and mortality of complications related to hypertension and diabetes by increasing screening and adherence to treatment guidelines [14]. Another study indicated that implementing an EHR improved hypertension and diabetes metrics, as well as patient satisfaction with clinical care [13]. Compared to a paper-based system, the reasons cited for improved health outcomes included increased recording of chronic disease metrics, enhanced history-taking, and more frequent lifestyle counseling, facilitated by EHR prompts [8,13,14]. Based on an analysis of various research efforts focused on EHRs, it can be concluded that EHRs have a positive impact on chronic disease outcomes by increasing provider adherence to guidelines and treatment algorithms, monitoring disease indicators, prompting counseling, and improving patient adherence [8,9].

An extensive review of the literature showed that there are few existing EHR systems that have been adopted by various health organizations to respond to the needs of vulnerable populations [8,15]. Each of these systems has its limitations for effective use and deployment, including high cost, restricted customizability across different clinical settings, the necessity of internet connectivity, and ongoing input from software engineers for successful deployment [15]. Therefore, there is still a need for an adaptable, equitable, and accessible EHR system.

Using a human-centered design approach, Hikma Health (HH), a US 501(c)(3) non-profit organization, has created a free, open-source EHR solution specifically for low-resource settings [6,7]. In prior research, we describe the user-centered design approach taken to develop this system with modular workflows and multiple languages [7]. The HH EHR system has demonstrated improved care continuity, clinical data visualization, and efficiency at clinics in Lebanon and Nicaragua, positively impacting the healthcare of more than 26,000 patients [6]. Even so, there is a need to improve and optimize the HH EHR system for higher functionality across different locations and clinical settings. Using a mixed-methods approach, this study analyzes the feasibility of the HH EHR system through its acceptability, practicality, ease of integration, and effectiveness [16,17]. in two different clinical settings in Lebanon and Nicaragua in order to direct further revision and implementation. Each clinic used a specific version of the HH EHR that was coded for the clinic's needs, and in the local language spoken at each clinic.

## Methods

### Study method overview

The study used a mixed-methods approach to obtain a suite of information on each organization and the experiences of its practitioners in using the HH EHR system. We based the design of this feasibility study on four key principles: acceptability, practicality, ease of integration, and effectiveness [16,17]. We selected these four principles since they were the most applicable to our specific innovation and implementation context [16,17]. Before our study began, we constructed definitions for these four principles based on prior literature [16,17].

First, a clinical demographic survey was given to identified administrators to better understand the context in which each organization provided care. Next, a purposive sampling of diverse healthcare providers in each organization was identified to obtain perspectives on the HH EHR. Each of these providers completed a sociodemographic questionnaire developed to understand the unique backgrounds that inform their clinical experience. Then, interviews were conducted with the providers to gain a deeper understanding of their practical experience using the HH EHR system.

### Clinical characteristics data collection

The clinical demographic survey was developed to better understand the context of each clinic. The survey was created in REDCap and sent to administrators via email at the beginning and end of the study in both EMA and NVC. Administrators were identified through previous communications with each clinic. Several sections were explored in the survey, including resources, staff, patients, diseases seen, and costs associated with maintaining the HH EHR system. These demographic

surveys were collected to understand the basic overall trends in patients seen and resources used in order to compare the contexts of each clinic and other data points.

### Qualitative data collection

In-depth interviews were conducted with both NVC and EMA to develop a deeper understanding of the feasibility of the HH EHR, particularly along the four categories of acceptability, practicality, ease of integration, and effectiveness. Acceptability was defined as how participants reacted to the HH EHR as either being suitable or challenging to learn or use. Practicality was defined by how participants found the HH EHR software, the hardware needed to operate the HH EHR, and the resources needed to utilize the HH EHR such as electricity and network connection. Integration was defined as how the HH EHR was integrated into each organization's workflow and patient-providers interactions. Efficacy was defined as how the HH EHR either positively or negatively affected clinical or patient outcomes such as clinical efficiency, clerical time, medical errors, and overall patient health outcomes.

For both locations, an interview guide was developed in an iterative fashion by four researchers (HA, ASC, SA, EN) trained in qualitative methods. This interview guide had sections for each theme: acceptability, practicality, ease of integration, and effectiveness (S1 Appendix). Interviews were conducted until thematic saturation was reached or all providers were interviewed. Healthcare workers with at least six months of experience using the HH EHR were considered; there were no other inclusion or exclusion criteria. Before each interview, each respondent was informed of the study standards and protocol, and written consent was obtained.

HA and ASC conducted in-person interviews at NVC from May 13th to May 22nd, 2022. Interviews were conducted face-to-face in Spanish in a private room at the clinic and continued until thematic saturation was reached. SA conducted interviews in Lebanon with EMA over Zoom from July 20th through August 30th, 2022. Interviews were conducted in either English or Arabic depending upon the respondent's preference, and all providers at EMA were interviewed.

A sociodemographic questionnaire was developed to understand the interview participants' background, training, and experience (S2 Appendix). This questionnaire was entered electronically by study personnel with each interviewee prior to conducting the interviews. To ensure data security and prevent a breach of confidentiality, the data were collected and stored in REDcap.

### Data analysis

Qualitative data from the in-depth interviews was analyzed using a Framework Analysis to identify themes and construct an initial codebook (HA, ASC). Four interviews were used to test and revise the codebook. The final codebook was applied to all interviews and coded by HA, TS, and YE. Data were transcribed, coded, and analyzed using NVivo (QSR International Pty Ltd. 2020). Inter-rater reliability of the coders was assessed using a Kappa coefficient.

Quantitative data from the clinic demographics surveys and specific quantitative questions in the in-depth interviews were analyzed in Microsoft Excel (Microsoft, Redmond, WA, USA). Standard descriptive analyses (mean, median, and ranges) of the clinical characteristics were performed.

### Ethical considerations

Research was conducted in principles embodied in the Declaration of Helsinki. Ethical approval was obtained from Mass General Brigham (IRB #2021P003407) and the American University of Beirut to conduct this study (IRB #SBS-2021–0426). No respondents withdrew from the study and there were no reported breaches of data security.

## Results

### Clinical characteristics

At the beginning and end of the study, a clinical characteristic survey was sent to both EMA and NVC to understand the specific context in which the HH EHR was being deployed. These characteristics can be seen in Table 1. Both EMA and

**Table 1. Clinical characteristics.**

|  | Nueva Vida Clinic (NVC) | Endless Medical Advantage (EMA) |
|---|---|---|
| Resources |  |  |
| **Number of devices** | 15 | 4 |
| **Number of devices broken** | 0 | 0 |
| **Hours without Wi-Fi a week** | 2 | 40 |
| **How much do you spend on the internet a month? (USD)** | 120 | 0 |
| Patients |  |  |
| **How many are seen in a month?** | ~3,500 | ~2,500 |
| **How many are seen in a year?** | 42,670 | 21,906 |
| **What percentage of patients have chronic diseases?** | 42% | 17% |
| **What percentage of patients have infectious diseases?** | 28% | 83% |
| **What percentage of patients are pregnant?** | 8% | 12% |

NVC had never used an electronic health record before the HH HER. However, there are some important differences between EMA and NVC that likely affected their deployment and use of the HH EHR. Overall, NVC was slightly larger, seeing about 1,000 more patients a month and approximately 20,000 more patients a year and 20% more patients with chronic diseases.. The higher volume likely stressed the HH EHR more than at EMA. Additionally, EMA had significantly more hours a week without access to Wi-Fi a week. Combined, differences likely impacted subsequent interviews as described below.

## Feasibility data analysis

Overall, eleven in-depth interviews were completed, including three nurses, six doctors, one dentist, and one pharmacist. Of the interviewees, six were women and five were men. The average experience of the interviewees was seven years and time using the HH EHR was eight months. There was substantial agreement in the qualitative framework analysis with a kappa of 0.76 between the three coders indicating strong agreement. During the in-depth interviews, participants were asked selective quantitative questions, the results of these questions can be seen in Table 2.

## Acceptability

Each respondent was trained through two to three personal one-hour sessions on how to use the HH EHR by an organization champion who has been trained by HH on how to use the EHR. Eight out of the eleven respondents thought this was sufficient training, while three thought there should have been more initial sessions. The average time reported to becoming comfortable with using the HH EHR was two and a half weeks, with a normal distribution except for one outlier that thought it took about 8 weeks to feel comfortable. Overall, younger users felt more comfortable than older users, with the three oldest users having the longest times to feel comfortable using the system. Entering patient data and using the modular workflows was reportedly easy to learn across both sites, but respondents at NVC felt it was harder to learn how to find specific data from other specialty clinics within the system.

**Table 2. Quantitative data analysis.**

|  | Nueva Vida Clinic (N=8) | Endless Medical Advantage (N=3) | Total |
|---|---|---|---|
| **Average number of hours trained** | 3.4 hours | 2.8 hours | 3.1 hours |
| **Average time need to feel comfortable using the HH EHR System** | 3.1 weeks | 2.1 weeks | 2.6 weeks |
| **Average Time the HH EHR changed to patient encounters** | -1.3 minutes | -4.8 minutes | -3.1 minutes |

Ten out of eleven respondents reported that the HH EHR was easy to use, and eight reported difficulties with using the HH EHR. Of the eight respondents noting difficulties, the most commonly reported difficulties were issues with using the search functionality to find patients across multiple transliterated spellings of a name, and issues with consistent syncing of the system. We address issues with syncing in the Practicality section below as a part of software challenges. Participants found that different phonetic and cultural ways of spelling names led to issues with searching for patient records and the creation of duplicate patient records. Below a doctor at NVC describes this dilemma.

*There were a lot of differences especially in pronunciation and transcription. For example, while entering the patient's name, each one can enter it in his own way, how he heard it, or transcribe it; there was no standard in the Hikma Health EHR that shows how this name should be written.*

This issue was partially addressed by the addition of patient photos, but not every patient wanted their photo taken and these photos increased the overall digital size of patient files.

Respondents at both sites noted that the HH EHR's simplified modular workflows decreased the amount of writing that users had to do to effectively complete a patient's encounter. Respondents at both sites also reported that the drop-down menus within each workflow decreased the time they spent typing and simplified the flow of a patient encounter. Four of the eleven respondents thought that the drop-down menus and standardized forms could be expanded to include the spelling of names and medications to decrease spelling errors and improve data searching in the HH EHR through standardization.

## Practicality

Nine out of eleven participants found the software practical, and all eleven found the hardware and resources needed to utilize the HH EHR practical for their respective settings.

The participants at the NVC experienced greater challenges with the use of the HH EHR due to their use of the system in a semi-online manner, with more continuous syncing throughout a patient's encounter with multiple clinicians. All participants from the NVC site mentioned challenges with synchronization. One nurse at NVC noted that:

*I didn't have a very good* [Wi-fi] *signal in my area. So, for me it was difficult to synchronize. I wrote my* [physical] *exams, they were saved, but they were not uploaded to the cloud. So, the other doctors couldn't see my results. It's like I was running behind, because I had to wait a long time for there to be a signal for it to synchronize.*

This issue was addressed initially by having a hybrid paper and electronic system, which created additional documentation responsibilities for clinicians. A longer-term resolution to this challenge was achieved by redesigning the architecture of the sync functionality of the system. Rather than syncing the entire database to and from every device during each sync, the system synced only new or edited events to the cloud server. This feature change eventually enabled the sync functionality to be more efficient and reliable when used by NVC clinicians in this semi-online manner. Because EMA used the system in an entirely offline-first capacity and did not need to sync at multiple points in a single patient encounter, they did not experience the same syncing challenges experienced by NVC.

All participants thought that the HH EHR worked well on both tablets and mobile phones; there were no reported issues with device maintenance, devices breaking, or malfunctioning. The only hardware issues reported was remembering to charge devices at EMA before trips were made to refugee camps. Overall, there was a preference for tablets among doctors who thought they were easier to type on.

Respondents also reported on the practicality of the resources needed to operate the HH EHR including electricity, Wi-Fi, and technical support. Overall, there were no challenges with any of these resources at EMA due to their operation as a remote clinic. As NVC required multiple syncs for a patient visit, they did experience issues with clinical flow if the

Wi-Fi was not working as seen in the quote in this prior section. Once synchronization issues were addressed, no additional technical support was needed in NVC.

### Integration

Clinical workflow improvements and difficulties were both mentioned in nine out of the eleven interviews. Almost all the clinical workflow difficulties were related to issues at NVC with synchronization. This is described well in a quote below from a NVC user describing how synchronization would not be fast enough and lead to miscommunications, delays in patient care, and frustration among staff:

> For example, the doctor synchronized the medicine that he wrote, and he is sure that that medicine has already arrived at the pharmacy, to the pharmacy device. And maybe the pharmacy device is not syncing. Then, the pharmacy guy comes and tells the doctor to please synchronize. And the doctor says that it has already synchronized. Then, the woman from the pharmacy comes and tells him that please write it on a sheet of paper, because the information has not reached them. Then, the doctor comes and tells him that he is not going to do double the work, because he already did it and already synced it. Then, it becomes chaos, both internally and with patients.

When issues with synchronization were fixed and the system did work well at NVC, it led to improvements in clinical efficiency and improvements in patient-provider interactions. In particular, the HH EHR was mentioned as being superior to the paper systems used previously at both EMA and NVC. One physician from NVC mentioned how previously, "20,000 [paper] files that we are removing every five years…" leading to a large loss in important patient data. With the HH EHR, providers also appreciated how patient data was summarized and easily understood. Seven participants mentioned how this improved provider-patient interactions and clinical visits times. We discuss more on clinical time below, but regarding provider-patient interactions, participants mentioned how with the HH EHR patients felt like "you were giving them personalized attention." Below is a quote from an EMA physician describing this interaction:

> Before we used to ask the patients so many questions about their last visit or their history to get at least 30% of the information, now while using this application [HH EHR] we can get all the information from their files, like their age, the chronic diseases, what we gave him as medication in his last visit. It is much easier to interact with the patient while using Hikma Health instead of asking him all these questions on every visit. Much better for saving time and information.

### Efficacy

From both quantitative and qualitative data, the HH EHR showed improvements in both clinical efficacy (improved patient outcomes, decreasing medical errors) and systems efficacy (clerical time, resource utilization).

All eleven participants reported that, when functioning, the HH EHR improved clinical efficacy. These clinical outcomes included: improved tracking of medications across patient visits, accuracy of patient diagnosis, and patient treatment. Participants attributed this to having improved patient information, both past medical history and notes from other physicians, which allowed for more accurate medical decisions. When asked to describe these improvements in clinical outcomes, a physician at EMA described the following:

> Yes, of course, it [patient outcomes] improved since Hikma Health helped us to reach very useful information. For example, we found that most of the women that are giving birth to abnormal children have a deficiency in folic acid. They are not taking folic acid or did not know that they will get pregnant. We were able to reach some research results

*while using Hikma Health, for example, what is the cause behind this disease, what is its percentage, and which ages are the most affected by this disease. We were able to find solutions for several diseases.*

Additionally, both sites reported a decrease in medical errors. One physician at NVC noted that before the HH EHR, "Sometimes, we wrote the dosage of the medicines incorrectly, or we had horrible handwriting and the person that was in the pharmacy had to guess." After implementing the HH EHR, both sites saw a large decrease in medical errors related to prescription medications since handwritten prescriptions were replaced with standardized drop-down menu categories.

One of the main improvements in the system's efficacy was the overall time clinicians had to spend recording patient information. Because clinicians had access to medical history that had previously been recorded and the simplicity of recording new data, users on average reported saving of about three minutes per patient interaction. One physician at EMA reported that consistently the HH EHR, "saved at least five minutes for each patient." Neither site reported large changes in resource utilization. NVC reported being better able to track which drugs were being prescribed, but they have not been able to make systems-level decisions with this data yet.

## Discussion

Due to the rise of conflict and climate change, millions of people are being forcibly displaced, creating new health challenges and strains existing health systems, calling for novel innovations [7,15,18–20]. Particularly as we have seen from new disasters such as that in Gaza, there is a need to provide healthcare in extremely low resource settings often without basics such as electricity or network access [21]. In this study we tested the feasibility of the first free, open-sourced, offline capable EHR specifically designed to care for vulnerable populations.

Overall, the HH EHR was deemed a feasible intervention based on its acceptability, practicality, integration and efficacy, but had limitations based on the context in which it is deployed. In comparing the interventions between NVC and EMA, the HH EHR had technical difficulties at NVC when it was synced multiple times during a patient visit as they moved from intake to the physician, and finally to the pharmacy. This is likely due to the fact that each sync would be about 500 megabytes, with multiple devices syncing, which strained the network connection. Additionally, if the network signal was weak or unavailable, it would be extremely challenging to sync between providers in a patient's clinical encounter. When this happened at NVC documentation became twice as challenging as providers would document in the HH EHR and also on paper. This not only burdens providers but also leads to a higher risk of medical errors or miscommunication. Issues with syncing at NVC led HH to improve the efficiency of the syncing functionality by adapting the system architecture to check for any amended or new data files and only syncing those rather than the entirety of the database each time. Thus, creating a more efficient experience while still being an offline-first EHR. These issues were not seen at EMA where the HH EHR was used with its offline-first capable focus. At EMA synchronization happening either once or twice a day once providers returned from remote refugee camps without internet access. These findings indicate that the implementation of an appropriate EHR solution greatly depends on the clinical context and the availability of resources.

This study adds to the growing body of literature that shows how EHRs are feasible and can improve health outcomes in resource limited settings for displaced and rural populations. In particular, this study is the first of its kind to demonstrate that offline-first EHRs are feasible. This finding is important for expanding the clinical settings in which EHRs are deployed, particular for displaced populations. Our findings agree with Khader et al. and Doocy et al. that have shown how EHRs can improve patient care through reliable longitudinal documentation, allowing for providers to titrate medicines and decrease time documenting during a clinical encounter [13,14,22]. Previous literature has discussed how one of the largest barriers to implementing EHRs in these contexts was provider uptake and use [8]. We found that the use of modular workflows and drop down menus decreased the need for providers to type, improving efficiency and also decreasing spelling errors. This innovation led to the HH EHR being more acceptable and practical to providers, with providers in the in-depth interviews asking for the addition of more drop-down menus across the platform.

## Limitations

There are a few important limitations in this study. First, it is important to note that this study assessed feasibility and some measures of effectiveness but was not a complete effectiveness study. We were able to find some measures of described effectiveness through our in-depth interviews, but our quantitative data was not rigorously tested to measure effectiveness. While our study was able to assess two sites, there were not more providers to interview at EMA as other physicians rotate through the clinic and do not stay longer than 6 months, meeting our exclusion criteria. However, across all the interviews thematic saturation was reached.

Several biases could have affected the results of this study. First, Hikma Health's sponsorship of this study leading to the influx of resources into respective clinics could have affected respondents' answers. While purposive sampling was used to ensure diverse perspectives were included, including non-clinical staff, this method may have introduced selection bias. Additionally, social desirability bias could have affected respondents' answers. To mitigate these possible biases, respondents were asked open ended questions and questions that were balanced with positive and negative experiences with the HH EHR.

## Conclusion

There is an urgent need to implement feasible solutions that meet the health needs of displaced and vulnerable populations, particularly as these groups grow from the rise in conflict and climate-related disasters. From the growing body of research, EHRs offer a feasible solution to improve the healthcare provided to populations in lower-resourced settings. Our study demonstrated that offline-first EHRs are feasible in very limited resource settings through improved documentation, leading to increased clinical efficiency and patient care. These findings are critical to expanding deployment of EHRs in settings that did not previously did not have the capabilities to utilize EHRs. It is our hope that with expanded deployment, patients around the world in any clinical setting can receive improved healthcare through offline-first EHRs.

## Supporting information

**S1 Appendix. Standard feasibility definitions.** This appendix contains all of the definitions used in our feasibility study. These definitions were built off standard definitions in the literature.
(DOCX)

**S2 Appendix. Interview guide for Hikma Health's Electronic Health Record (EHR) project.** This appendix has the interview guide that was used to conduct the in-depth interviews with healthcare workers and administrators.
(DOCX)

**S3 Appendix. Hikma health EHR background questionnaire.** This questionnaire was given to every in-depth interview participant to understand their experience that could have informed their interview.
(DOCX)

**S4 Appendix. SQUIRE checklist.** This is the Standard for Quality Improvement Reporting Excellence (SQUIRE 2.0) checklist, used to ensure that this study adhered to the standards of mixed-methods studies.
(DOCX)

## Author contributions

**Conceptualization:** Henry Ashista, Samar Al-Hajj, Erica Nelson.

**Data curation:** Henry Ashista, Alanis Santiago Comas, Samar Al-Hajj.

**Formal analysis:** Henry Ashista, Alanis Santiago Comas, Taylor Selby, Mohammad Yasir Essar, Jude Alawa, Erica Nelson.

 

**Investigation:** Henry Ashista, Alanis Santiago Comas, Taylor Selby, Mohammad Yasir Essar.

**Methodology:** Henry Ashista, Alanis Santiago Comas, Samar Al-Hajj, Erica Nelson.

**Project administration:** Henry Ashista.

**Resources:** Henry Ashista.

**Supervision:** Henry Ashista, Samar Al-Hajj, Erica Nelson.

**Validation:** Henry Ashista, Samar Al-Hajj, Erica Nelson.

**Writing – original draft:** Henry Ashista, Alanis Santiago Comas, Taylor Selby, Jude Alawa, Erica Nelson.

**Writing – review & editing:** Henry Ashista, Taylor Selby, Mohammad Yasir Essar, Jude Alawa, Samar Al-Hajj, Erica Nelson.

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
