## [Decision Letter · Decision Letter 0]

29 Jul 2025

Response to Reviewers
Revised Manuscript with Track Changes
Manuscript
**Journal Requirements:**
**Additional Editor Comments (if provided):**
**Reviewers' Comments:**

**Comments to the Author**

1. Does this manuscript meet PLOS Digital Health’s publication criteria?

Reviewer #1: Yes

Reviewer #2: Partly

2. Has the statistical analysis been performed appropriately and rigorously?

Reviewer #1: I don't know

Reviewer #2: N/A

3. Have the authors made all data underlying the findings in their manuscript fully available (please refer to the Data Availability Statement at the start of the manuscript PDF file)?

Reviewer #1: Yes

Reviewer #2: Yes

4. Is the manuscript presented in an intelligible fashion and written in standard English?

Reviewer #1: Yes

Reviewer #2: Yes

Reviewer #1: Please see attached comments regarding possible options to consider re: bias and limitations and next steps for future considerations. I have also included some elements regarding health consumer literacy and elements re data sovereignity challenges used in vulnerable populations globally.

Reviewer #2: Major Issues

1. Lack of thematic justification

The manuscript investigates the feasibility of the Hikma Health Electronic Health Record (HH EHR) across four categories: acceptability, practicality, ease of integration, and effectiveness. However, there is no clear justification provided for the selection of these specific themes. The authors should explain why these dimensions were chosen over others, and whether they are based on an established framework, previous literature, or theoretical rationale. Without this, the thematic focus appears arbitrary and limits the interpretability or relevance of the findings.

2. Contributing factors relevant to clinical and technological characteristics

Another significant omission is the lack of discussion on how previous clinical characteristics, as presented in Table 1, may have influenced participants’ perceptions of the HH EHR. These variables could meaningfully impact user experience and adoption. Additionally, the study does not consider participants’ baseline technological familiarity or digital literacy. Excluding these considerations undermines the validity of the conclusions, particularly those related to ease of use and acceptability

3. Limited discussion on offline EHR system benefits

The discussion does not sufficiently consider the potential advantages of using an offline HH EHR system, even if it lacks the full capabilities of the online solutions. While the limitations of offline functionality are noted, it would strengthen the manuscript to also recognise that offline systems may still provide meaningful benefits, such as improved documentation and continuity of care. Including this perspective would offer a more balanced justification for the implementation of such solutions, especially in low-resource or connectivity-constrained settings.

**Do you want your identity to be public for this peer review?** For information about this choice, including consent withdrawal, please see our Privacy Policy

Reviewer #1: No

Reviewer #2: **Yes:** Noushin Nazarian

**Figure resubmission:****Reproducibility:** To enhance the reproducibility of your results, we recommend that authors of applicable studies deposit laboratory protocols in protocols.io, where a protocol can be assigned its own identifier (DOI) such that it can be cited independently in the future. Additionally, PLOS ONE offers an option to publish peer-reviewed clinical study protocols. Read more information on sharing protocols at https://plos.org/protocols?utm_medium=editorial-email&utm_source=authorletters&utm_campaign=protocols

---

## [Decision Letter · Decision Letter 1]

18 Nov 2025

Response to Reviewers
Revised Manuscript with Track Changes
Manuscript
**Journal Requirements:**
**Additional Editor Comments (if provided):**
**Reviewers' Comments:**

**Comments to the Author**

Reviewer #2: (No Response)

publication criteria?

Reviewer #2: Yes

3. Has the statistical analysis been performed appropriately and rigorously?

Reviewer #2: N/A

4. Have the authors made all data underlying the findings in their manuscript fully available (please refer to the Data Availability Statement at the start of the manuscript PDF file)?

Reviewer #2: Yes

5. Is the manuscript presented in an intelligible fashion and written in standard English?

Reviewer #2: Yes

Reviewer #2: in response to second major issue raised, you review highlights the response are within Page 12, Lines 230-232, but no relevant information are found in this page or within those lines

**Do you want your identity to be public for this peer review?** For information about this choice, including consent withdrawal, please see our Privacy Policy

Reviewer #2: No

**Figure resubmission:**

**Reproducibility:** To enhance the reproducibility of your results, we recommend that authors of applicable studies deposit laboratory protocols in protocols.io, where a protocol can be assigned its own identifier (DOI) such that it can be cited independently in the future. Additionally, PLOS ONE offers an option to publish peer-reviewed clinical study protocols. Read more information on sharing protocols at https://plos.org/protocols?utm_medium=editorial-email&utm_source=authorletters&utm_campaign=protocols

---

## [Decision Letter · Decision Letter 2]

8 Jan 2026

An offline-first electronic health record for vulnerable populations: a mixed-methods feasibility

PDIG-D-25-00116R2

Dear Dr. Ashista,

We are pleased to inform you that your manuscript 'An offline-first electronic health record for vulnerable populations: a mixed-methods feasibility' has been provisionally accepted for publication in PLOS Digital Health.

Best regards,

Leo Anthony Celi, MD MS MPH

Editor-In-Chief

PLOS Digital Health

**Additional Editor Comments (if provided):**

**Reviewer Comments (if any, and for reference):**

Reviewer's Responses to Questions

**Comments to the Author**

Reviewer #2: All comments have been addressed

publication criteria?

Reviewer #2: Yes

3. Has the statistical analysis been performed appropriately and rigorously?

Reviewer #2: N/A

4. Have the authors made all data underlying the findings in their manuscript fully available (please refer to the Data Availability Statement at the start of the manuscript PDF file)?

Reviewer #2: Yes

5. Is the manuscript presented in an intelligible fashion and written in standard English?

Reviewer #2: Yes

Reviewer #2: NA

**Do you want your identity to be public for this peer review?** For information about this choice, including consent withdrawal, please see our Privacy Policy

Reviewer #2: No
